# $k$NN-LM Does Not Improve Open-ended Text Generation

**Shufan Wang**[1]  **Yixiao Song**[1]  **Andrew Drozdov**[1]
**Aparna Garimella**[2]  **Varun Manjunatha**[2]  **Mohit Iyyer**[1]
University of Massachusetts Amherst[1]    Adobe Research[2]
{shufanwang, yixiaosong, adrozdov, miyyer}@umass.edu
{garimell,vmanjuna}@adobe.com

## Abstract

In this paper, we study the generation quality of interpolation-based retrieval-augmented language models (LMs). These methods, best exemplified by the $k$NN-LM (Khandelwal et al., 2020), interpolate the LM's predicted distribution of the next word with a distribution formed from the most relevant retrievals for a given prefix. While the $k$NN-LM and related methods yield impressive decreases in *perplexity*, we discover that they do not exhibit corresponding improvements in *open-ended generation quality*, as measured by both automatic evaluation metrics (e.g., MAUVE) and human evaluations. Digging deeper, we find that interpolating with a retrieval distribution actually *increases* perplexity compared to the baseline LM for the majority of tokens in the WikiText-103 test set, even though the overall perplexity is lower due to a smaller number of tokens for which perplexity dramatically decreases after interpolation. However, when decoding a long sequence at inference time, significant improvements on this smaller subset of tokens are washed out by slightly worse predictions on most tokens. Furthermore, we discover that the entropy of the retrieval distribution increases faster than that of the base LM as the generated sequence becomes longer, which indicates that retrieval is less reliable when using model-generated text as queries (i.e., is subject to exposure bias). We hope that our analysis spurs future work on improved decoding algorithms and interpolation strategies for retrieval-augmented language models.

## 1  Introduction

Retrieval-augmented language models, which integrate non-parametric dense retrieval with autoregressive next-token prediction, have been validated with strong empirical performance across a variety of tasks (Metzler et al., 2022; Basu et al., 2022; Mialon et al., 2023) in addition to achieving low held-out perplexities on LM benchmarks. In this

paper, we study *interpolation-based* LMs, a subtype of retrieval-augmented LMs that compute the probability of the next token by interpolating between the softmax distribution of the original LM and a token distribution formed by retrieving over an external datastore. These methods, perhaps best exemplified by the $k$NN-LM (Khandelwal et al., 2020), are particularly attractive because they allow any pretrained LM to be retrofitted with a retrieval module without further training.

Despite these advantages, there is limited understanding about the *text generation quality* of interpolation-based LMs. In this study, we evaluate the quality of generated text from two such methods, $k$NN-LM and TRIME (Zhong et al., 2022), against the output of baseline LMs that do not use retrieval. Our evaluations involves *open-ended* text completions generated using different decoding algorithms on both the WikiText-103 and PG-19 datasets. We discover that interpolation-based LMs do not improve the quality of generated text, as measured by both automatic text generation metrics such as MAUVE (Pillutla et al., 2021) and human evaluation.

This result begs the question of *why* the text generation quality does not improve, as the perplexity of interpolation-based LMs is substantially lower than that of the baselines. Our analysis of the $k$NN-LM model suggests two potential reasons for this lack of improvement:

1. $k$NN-LM actually *worsens* the predictions of the majority of tokens in the WikiText-103 test set. On aggregate, perplexity improves because of significantly improved predictions on a smaller subset of tokens. However, when generating a long sequence of tokens, these improvements are washed out by the worsened predictions on other tokens.

2. The quality of the retrieval distribution deteriorates faster than that of the LM's predicted

distribution as the length of the generation increases; in other words, the retrieval distribution is more vulnerable to exposure bias and can be easily thrown off by artifacts presented in model-generated text.

Unlike previous works that rely on perplexity to evaluate language modeling or BLEU to evaluate machine translation quality of $k$NN-LM-based models (Khandelwal et al., 2021), our work specifically studies the open-ended text generation capability of $k$NN-LMs with a range of automatic evaluation metrics as well as human evaluation. We demonstrate that, though they significantly lower perplexity, retrievers might also impair text generation performance of $k$NN-LMs. This finding suggests potential future directions for using retrieval during text generation, such as developing more robust retrieval components or employing retriever mechanisms more selectively during decoding.

## 2    Related Work

We present the most extensive study of open-ended text generation[1] from interpolation-based LMs such as $k$NN-LM (Khandelwal et al., 2020). Our results reveal that although these methods are effective at reducing perplexity, they can also be detrimental to text generation. Previous work finds that retrieval LMs are improved by selectively incorporating retrieval when conditions are favorable (He et al., 2021a; Alon et al., 2022; Drozdov et al., 2022; Mallen et al., 2023), although they only examine the teacher-forced setting or other tasks, e.g. question answering. The $k$NN-MT (Khandelwal et al., 2021) explores machine translation, which is a constrained task with short inputs, and thus not a good test of open-ended long-form generation.

The $k$NN-LM effectively scales retrieval to billions of tokens using a token-level non-parametric interpolation technique first introduced by Grave et al. (2017). Alternative retrieval-augmented models experiment with training the retriever (Zhong et al., 2022; Ram et al., 2023; Shi et al., 2023), interpolating vectors instead of token probabilities (Yogatama et al., 2021), scaling to trillions of tokens (Borgeaud et al., 2021), exploiting retrieval for strong few-shot learning (Izacard et al., 2022), and so on (Chen et al., 2017; Guu et al., 2020; Lewis et al., 2020; Izacard and Grave, 2021; Rae

et al., 2021; Wu et al., 2022; Trivedi et al., 2022; He et al., 2022). Among these, $k$NN-LM stands out as a relatively simple and fundamental work. Our findings indicate important weaknesses of retrieval for text generation.

Reference-based metrics are not well suited to evaluate open-ended text generation (Novikova et al., 2017). Instead, effective automated approaches compare the machine generated and human language text distributions using samples (McCoy et al., 2021; Pillutla et al., 2021; Pimentel et al., 2023). Human evaluation remains the golden standard for natural language generation (Hashimoto et al., 2019; Celikyilmaz et al., 2020; Krishna et al., 2023).

## 3    Experimental setup

Using a variety of commonly used text generation evaluation metrics, we evaluate the text generation capability of interpolation-based LMs and compare them to baseline LMs (i.e., without $k$-nearest-neighbor retrieval from an external datastore). In this section, we describe our experimental setup, including models, automatic evaluation metrics, data selection, and hyperparameters.

### 3.1    Models

We experiment with two interpolation-based LMs: the $k$NN-LM of Khandelwal et al. (2020), which augments an existing pre-trained LM with a retrieval module without any additional training, and TRIME (Zhong et al., 2022), a recent improvement over the $k$NN-LM that trains the retriever and LM jointly to further decrease perplexity.

$k$**NN-LM:**    The $k$NN-LM is a pre-trained language model that uses retrieval to improve word prediction. We follow the procedure from Khandelwal et al. (2020) and Alon et al. (2022)[2], and use the LM to encode token-level representations from a document collection (e.g., WikiText-103 training data) into a datastore where each token in document is converted into a key-value pair: a context vector $k_i$ representing the first $n-1$ words and a value $v_i$ which is the $n$-th word. During evaluation, the model calculates Euclidean distances $d(k, q_j)$ between the query vector $q_j$ and all the keys $k_1, k_2, \ldots k_{|V|}$ in the datastore. The values

---

[1]The $k$NN-LM is also evaluated using MAUVE in Lan et al. (2023); however, our work has much more extensive analysis in the open-ended text generation setting.

[2]Alternative architecture options for $k$NN-LM are explored in Xu et al. (2023). We don't expect those settings to impact the trends we observe, but as we mention in §6, tuning for text generation could be beneficial.

from the retrieved documents define a new distribution of the next word:

$$P_{KNN}(w_t|q_t) \propto \sum_{(k_i, v_i)} \mathbb{1}_{w_t = v_i} \exp(-d(k_i, q_t))$$

The model interpolates the LM's predicted distribution over the next token $P(w_t|q_t)$ with the retrieval distribution with a tunable hyperparameter $\lambda$:

$$P'(w_t|q_t) = \lambda P_{KNN}(w_t|q_t) + (1-\lambda)P_{LM}(w_t|q_t) \quad (1)$$

To generate text from the $k$NN-LM, we apply a decoding strategy (e.g., greedy decoding or truncated sampling algorithms) using the final interpolated probability distribution $P'(w_t|q_t)$.

**TRIME:** Note that in $k$NN-LM, the LM is trained *without* retrieval; the retrieval component is bolted on after training. Zhong et al. (2022) note that this approach is suboptimal, as the LM does not understand how to best use the retrieval. Thus, they propose TRIME, which uses an efficient in-batch strategy to incorporate retrievals during training. While $k$NN-LM relies on just one type of retrieval (from an external datastore), TRIME can retrieve from local, long-range, as well as external context. We use the TRIME$_{\text{EXT}}$ configuration in all of our experiments, which also uses a linear interpolation between LM and retrieval distributions (as in Equation 1) to produce the final probability distribution. The baseline LM (no external retrieval) retrieves from example-level local and long context, but has no access to a huge-scale external datastore.

### 3.2 Constructing an evaluation dataset

We sample from WikiText-103 (Merity et al., 2016) to construct our main evaluation dataset; in Section 4, we also perform an analysis experiment on the PG-19 dataset (fictional books) to test whether our findings hold across domains. We choose WikiText-103 because it is the most commonly used dataset for evaluating interpolation-based LMs; indeed, the main experiments from both $k$NN-LM and TRIME demonstrate that the retrieval component decreases held-out perplexity on this dataset compared to the baseline LM. Specifically, we randomly sample 5K examples[3] from the

validation set of WikiText-103. [4]

### 3.3 Automatic evaluation metrics

For all models tested, we compare the quality of text generated by the baseline LM with and without the $k$-NN retrieval component over the external datastore. We measure quality via the following automatic metrics:

**MAUVE:** MAUVE is an evaluation metric for open-ended text generation (Pillutla et al., 2021) that achieves high correlation with human judgments of text quality. It measures the distribution similarity between the generated text and the reference text. Higher MAUVE scores indicate closer distance between the distribution of the generated text and that of reference text.

**RankGen:** Given a prefix and several possible continuations (suffixes), RankGen (Krishna et al., 2022) outputs a score for each suffix, measuring the relevance between the prefix and suffix. Higher RankGen scores indicate stronger relevance between generated suffix with the given prefix. We thus measure the RankGen score between prefix and generated suffix for each of the two models.

**GPT-3 perplexity:** We use GPT-3 (Brown et al., 2020),[5] a large-scale pre-trained language model, to compute the perplexity of text generated with and without interpolation conditioned on the same prefix. Lower GPT-3 perplexity indicates stronger relevance between prefix and generated suffix and the better fluency of the generated suffix.

**Entity-F1:** Previous works (Nan et al., 2021; Lee et al., 2022) use the percentage of hallucinated named entities (entities that appear in the generated text but not in the reference text) or the ratio of named entity overlaps between the generated text and reference text to estimate the factuality of the generated text. In our work, we compute the F1 scores between the named entities from the generated text and reference text as a proxy for entity hallucination. Higher F1 scores may correlate to fewer instances of hallucinated entities.

**Seq-Rep-1:** We follow Welleck et al. (2020) and use the percentage of unique unigrams (Seq-Rep-1)

---

[3]We choose 5K examples because this is the minimum recommended number of generations to obtain meaningful comparisons as per Pillutla et al. (2021).

[4]We use the first 128 tokens of each example as a *prefix* that the model must condition on to generate a 256-token-long continuation. As some of our metrics requires reference text, we also store the ground-truth 256 tokens (*gold suffix*) that follow the prefix in each example.

[5]We use the 6.7B `gpt3-curie` model via OpenAI's API

in the text as a metric for lexical diversity in the text. Higher Seq-Rep-1 scores indicate lower diversity (more repetition) in the generated text.[6]

### 3.4 Model configurations and hyperparameters

In this work, we leverage pretrained model and datastore checkpoints released by prior work, and also train our own interpolation-based LMs.

**Baseline LM details:** For $k$NN-LM, we use the implementations from Alon et al. (2022) and Khandelwal et al. (2020). The model in Alon et al. (2022) relies on a backbone 117M-parameter GPT-2 small model (Radford et al., 2019) fine-tuned on the WikiText-103 training data. The external datastore is constructed by the same backbone model, and both the pretrained LM and datastore are publicly released by Alon et al. (2022).[7] We also test the model in Khandelwal et al. (2020), which proposes the first $k$NN-LM. Khandelwal et al. (2020) uses a 247M-parameter Transformer LM trained from scratch on WikiText-103 and the datastore is computed using the trained Transformer LM. For TRIME, we adopt the 247M-parameter TRIME$_{ext}$ model trained from scratch on WikiText-103 and publicly released by Zhong et al. (2022). Our "non-retrieval" baseline is the same model without external retrieval; in other words, it has access to only the local memory (recent tokens) and long-range memory (in-batch tokens). In all three set-ups, the external datastore is constructed using the training dataset of WikiText-103; the datastores from Zhong et al. (2022) and Khandelwal et al. (2020) both have 103M entries, while the datastore from Alon et al. (2022) has 117M entries (the discrepancy is due to tokenization differences between the models).

**Perplexity improvements from retrieval:** All models studied in this paper substantially decrease perplexity on WikiText-103's validation set when interpolation is enabled. For the model in Alon et al. (2022), the base GPT-2 perplexity is 14.8, and it decreases to 12.6 (-2.2) after interpolation. The $k$NN-LM in (Khandelwal et al., 2020) decreases perplexity from 17.96 (no retrieval) to 16.06 (-1.9) after interpolation. Meanwhile, TRIME decreases

---

[6]We also compute Seq-Rep-$N$ for $N = 2, 3, 4$, and observes consistent results with using Seq-Rep-1 (in Appendix A.4).

[7]See the gpt2-finetuned-wikitext103 model available here: https://github.com/neulab/knn-transformers.

| Model | MAUVE↑ | PPL$_{GPT-3}$↓ | RankGen↑ | EntityF1↑ | SeqRep$_1$↓ |
|---|---|---|---|---|---|
| *kNN-LM with and without retrieval from Alon et al. (2022)* | | | | | |
| GPT-2 small *(no retrieval)* | 77.7 | 13.1 | 11.7 | 14.2 | 56.7 |
| GPT-2 small *(+ retrieval)* | 79.2 | 14.8 | 11.7 | 13.1 | 53.3 |
| *kNN-LM (Khandelwal et al., 2020) with and without external retrieval* | | | | | |
| Transformer *(no retrieval)* | 89.5 | 20.4 | 12.9 | 12.1 | 41.8 |
| Transformer *(+ retrieval)* | 90.7 | 28.9 | 12.5 | 9.77 | 37.9 |
| *TRIME$_{EXT}$ with and without external retrieval from Zhong et al. (2022)* | | | | | |
| TRIME *(no retrieval)* | 90.6 | 22.2 | 13.1 | 11.3 | 40.1 |
| TRIME *(+ retrieval)* | 87.3 | 23.8 | 12.5 | 9.80 | 38.5 |

Table 1: Automatic evaluation metrics do not show consistent improvement in generation quality for interpolation-based LMs compared to their non-retrieval baseline LMs. We evaluate three set-ups: 1) $k$NN-LM with GPT2 as the baseline (top), 2) the original $k$NN-LM proposed in (Khandelwal et al., 2020) which trains a Transformer LM from scratch on the WikiText-103 training data (middle), and 3) TRIME which trains both the LM and the retrieval mechanism (bottom).

perplexity from 17.0 (no retrieval) to 15.5 (-1.5) after interpolation.

**Hyperparameters:** To generate text, we use the hyperparameters recommended by the authors that yield low perplexities on the WikiText-103 validation set. For the model in Alon et al. (2022) and Khandelwal et al. (2020), the softmax temperature is set to $1.0$ and the interpolation coefficient between the LM distribution and the retrieval distribution $\lambda$ is set to $0.25$. For TRIME(Zhong et al., 2022), the softmax temperature is set to $1.25$ and the $\lambda$ is $0.3$. For most of our experiments (e.g., those in Table 1), unless otherwise specified, we use nucleus sampling (Holtzman et al., 2020) with $p = 0.8$ for text generation.

## 4 Results

We find that despite incorporating the retrieval component and interpolating information from the baseline LM and retrieval, these methods do not yield any significant improvement to text generation performance, and even worsen it by some metrics (Table 1). In this section, we provide an overview of our main results, perform more fine-grained analyses, and describe a human evaluation that supports the conclusions drawn from automatic metrics.

**Interpolation-based LMs do not improve automatic text generation evaluation metrics:** We find that none of the three models significantly improve generation quality compared to the baseline LM, as shown by various metrics (Table 1). For the model in Alon et al. (2022) (top row in Table 1), while the MAUVE score improves by 1.5 points with retrieval, the perplexity of GPT-3 *increases* on retrieval-augmented generations, and the RankGen score is identical. For the model in Khandelwal et al. (2020) (middle row in Table 1), retrievals improves the MAUVE score even less significantly (1.2 points) but worsens perplexity of GPT-3, RankGen and Entity-F1. For TRIME (bottom row in Table 1), the non-retrieval baseline is actually slightly *better* across MAUVE, GPT-3 perplexity, RankGen and Entity-F1 . In other words, there is no convincing winner; furthermore, contrary to the expectation that $k$NN-LMs reduce hallucination by retrieving (and potentially copying) from the datastore, we do not observe any improvement in the Entity F1 scores with the gold suffix. We observe a marginal improvement in lexical diversity of the generations (shown by the lower Seq-Rep-1 score [8]).

**These results hold across different decoding algorithms:** The results in Table 1 are all from nucleus sampling. What if we change the decoding algorithm? To investigate the impact of decoding algorithm on generation quality, we evaluate the $k$NN-LM on three different decoding algorithms: greedy decoding, top-$k$ sampling, and beam search. We observe in Table 2 that none of these decoding algorithms changes the result: there is no clear winner between models with and without retrieval.

**These results hold across different datasets:** In addition to WikiText-103, we also evaluate the text generation performance of the $k$NN-LM on the PG-19 dataset (Rae et al., 2020), which predominantly comprises fictional books and presents a distinct thematic variation to Wikipedia. We construct an evaluation dataset from PG-19 similarly to our constructed evaluation dataset from WikiText-103 in Section 3.2. [9] The baseline LM is GPT2-

| Model | Nucleus Sampling | Top-$k$ Sampling | Greedy Decoding |
|---|---|---|---|
| *kNN-LM with and without retrieval from Alon et al. (2022)* | | | |
| GPT-2 small *(no retrieval)* | 77.7 | 87.1 | 2.32 |
| GPT-2 small *(+ retrieval)* | 79.2 | 87.5 | 2.44 |

Table 2: The observation that $k$NN-LM does not significantly improve text generation performance (measured here via MAUVE) is consistent across a variety of decoding algorithms: nucleus sampling, top-$k$ sampling ($k = 40$) and greedy decoding. We note that beam search decoding often generates repetitive text and therefore scores poorly with MAUVE.

| Model | MAUVE↑ | PPL$_{GPT-3}$↓ | RankGen↑ | EntityF1↑ | SeqRep$_1$↓ |
|---|---|---|---|---|---|
| *kNN-LM with and without retrieval from PG-19 (Rae et al., 2019)* | | | | | |
| GPT-2 small *(no retrieval)* | 8.00 | 17.3 | 4.13 | 5.63 | 47.6 |
| GPT-2 small *(+ retrieval)* | 9.26 | 18.8 | 3.62 | 4.87 | 44.5 |

Table 3: Consistent with our findings in WikiText-103 dataset, we find in PG-19 (fictional books) that $k$NN-LM does not yield consistent improvement in text generation quality compared to no-retrieval baseline LMs.

small model fine-tuned on the PG-19 dataset for three epochs (with 28.9 perplexity on the validation dataset).[10] Table 3 shows that on the PG-19 dataset, $k$NN-LM also does not improve text generation quality. While (marginally) improving perplexity, the $k$NN-LM often returns unhelpful artifacts from the PG19 dataset (see examples in Appendix A.3).

## 4.1 Human evaluation

Having found that interpolation-based LMs do not notably improve text generation quality according to automatic evaluation metrics, we turn next to human evaluation, which is known to be more reliable for generation tasks (Celikyilmaz et al., 2020; Krishna et al., 2021), to compare the text generated by the $k$NN-LM vs. the baseline GPT-2 model from Alon et al. (2022). We hired three English teachers/editors on the freelance marketplace Upwork. The evaluation was conducted on the platform Label Studio (Tkachenko et al., 2020-2022).[11] The

---

[8] We also report the Seq-Rep-N scores for N=2, 3, 4 in Appendix A.4

[9] From the validation dataset of PG-19, we randomly sample 5K samples, where in each sample, the first 128 tokens is used as the *prefix*. For datastore construction, we sample 1536 books from the training dataset only (filtering out the first 10% and last 10% tokens of each books for irrelevant content such as copyright statements). Our training dataset

and datastore consist of 98M tokens, similar in size to those in the WikiText-103 dataset.

[10] Consistent with Drozdov et al. (2022), the model trained on PG-19 gives both worse MAUVE score and perplexity compared to the model trained on WikiText-103 since the PG-19 is a more diverse and challenging dataset.

[11] https://www.upwork.com, https://labelstud.io/

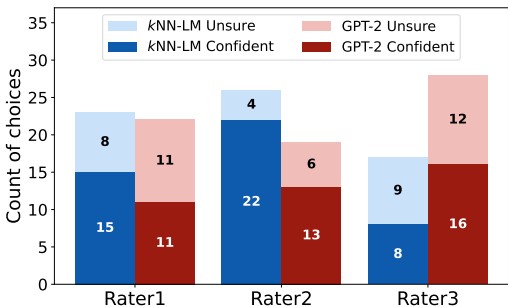

Figure 1: The plot presents how many times each type of generations ($k$NN-LM or GPT-2) is chosen by the evaluators. The dark area in each bar shows that the choices were made confidently. The light area represents the choices between $k$NN-LM and GPT-2 that were hard but the evaluator still chose the corresponding type. Overall, annotators preferred GPT-2 baseline texts 51% of the time compared to 49% for $k$NN-LM.

annotators were experienced in text generation evaluation and hired after careful selection.

The annotators were given a prefix and two continuations of the context (one generated by the baseline LM and one generated with retrieval, with randomized presentation order). The evaluators' task was to decide which continuation is better, indicate whether it was hard to choose between the two following Thai et al. (2022), and justify their choice in 3 to 4 sentences.[12] The evaluation focused on whether the generated text is grammatical, fluent, consistent, and logical.[13]

**Human evaluation shows no definitive winner between $k$NN-LM and GPT-2 either:** On aggregate, baseline GPT-2 generations were preferred 51% of the time, vs. 49% for $k$NN-LM. Additionally, the three annotators report that the decision was difficult for 37% of all cases. For Rater1 and Rater3, the rates of *difficult to choose* are as high as 42% and 47% while for Rater2 it is 22%. Out of the 45 comparison pairs, the three annotators only agree on their choices in 17 instances (37.78%), resulting in a Fleiss Kappa score 0.17 (slight agreement). Figure 1 presents the evaluator preference when comparing the $k$NN-LM to GPT-2 generations.

**Both models make catastrophic errors at similar rates:** A qualitative analysis of the the evaluators'

---

[12]A screenshot of our evaluation platform can be found in Appendix A.

[13]Each evaluator evaluated 45 pairs of continuations generated by $k$NN-LM and GPT-2. Each evaluator was paid $50 for their work.

justifications reveals that both $k$NN-LM and GPT-2 make catastrophic mistakes. Table 5 gives four examples of bad continuations, along with the evaluators' comments and our categorization of the errors. In the first row of the table, Continuation A generated by the $k$NN-LM contains repetitive content (i.e., ==ZAPU retreat==), and confuses *ZAPA* and *ZIPRA* at multiple places. The GPT-2 continuation in the second row states that a person was born in 1584 but was still alive in 1742; the generation in the third row by the $k$NN-LM claims that U.S. Route 75 curves both northeast and northwest in the northbound direction. Furthermore, both the GPT-2 and $k$NN-LM's generations change topics abruptly as shown in the lower half of Table 5. Overall, the quantitative and qualitative analyses of the human evaluation results show that the $k$NN-LM does not clearly improve over its base GPT-2 model despite its significant improvement in perplexity.

## 5 Why do $k$NN-LMs fail to improve text generation quality?

Our evaluations (both human and automatic) do not show a significant quality increase when interpolating an LM's predicted probability distribution with one formed via retrieval over large external datastores. In this section, we try to understand *why* we do not observe an improvement by empirically analyzing the $k$NN-LM and find two potential reasons: (1) despite lowering the aggregate perplexity, $k$NN-LMs only improve the perplexity of 42% of all test tokens, which suggests that the improved quality of a subset of tokens could be counter-balanced by worsened predictions on other tokens that do not benefit from the $k$NN-LM. Moreover, we find the entropy of the retrieval distribution to increase at a faster rate than that of the baseline LM as the model generates longer sequences. This difference implies that the retriever distribution is getting noisier as more tokens are sampled, potentially due to the exposure bias stemming from the retriever having to rely on the sampled text as the query.

### 5.1 KNN-LMs only benefits a subset of tokens

Many studies have shown that $k$NN-LMs decrease perplexity via retrieval interpolation (Khandelwal et al., 2020; Alon et al., 2022; Drozdov et al., 2022). Previous work (Drozdov et al., 2022; Zhong et al., 2022) has also suggested that $k$NN-LMs benefit the inference of tokens of various part-of-speech (POS) tags to different degrees (by lowering the perplexity

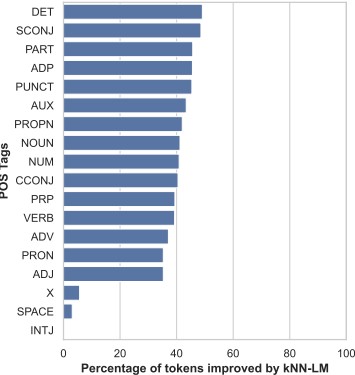

Figure 2: Across all POS tags, we observe that $k$NN-LM does not increase the probability of the majority of gold next token predictions. For verbs, pronouns, and adjectives, it only helps $< 40\%$ of the time (i.e., it hurts the predictions of the majority of these tokens).

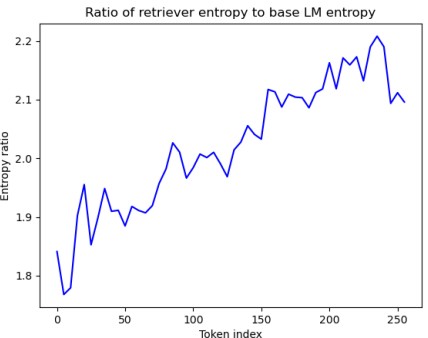

Figure 3: We plot the ratio between the Shannon entropy of the retriever's next-token distribution and that of the baseline LM softmax distribution, as the number of generated tokens increases. The ratio increases for longer model-generated sequences, indicating that the retriever becomes less confident than the baseline LM as decoding progresses.

of the gold token). However, these works focus on **aggregate** perplexity averaged across tokens in the test data but do not look at **individual** tokens and the percentage that actually benefit from retrieval.

Using the dataset we selected from WikiText-103, we compute the percentage of gold tokens from our test examples that are assigned lower perplexity (higher probability) by the $k$NN-LM compared to the base LM. We find that only 42% of the tokens benefit from $k$NN-LMs, while the remaining 58% of the tokens are adversely affected by the $k$NN-LM (i.e., the $k$NN-LM assigns a lower probability to the gold token compared to the base-LM). Moreover, we calculate the percentage of gold tokens that benefit from $k$NN-LM in each POS category (Figure 2) and consistently find the similar result that $k$NN-LM only helps reduce the perplexity for a smaller subset of tokens. We show examples of $k$NN-LM negatively impacting the next-token prediction (assigning the gold token with lower probability than the base-LM) in Table 4.

This means that despite lowering the **aggregate** perplexity across the test sets, the $k$NN-LM is more likely to hurt, instead of help, the inference of each **individual** token. Therefore, we hypothesize that during text generation, as the model samples a sequence of tokens, the advantages brought by $k$NN-LM to a smaller subset of tokens are offset by other tokens, for which $k$NN-LM may even have a detrimental impact on the inference.

## 5.2 The retriever becomes less reliable with longer generated sequences

Additionally, we observe that as the model generates longer sequences of text, the retriever component from $k$NN-LM becomes less confident and reliable in returning a high-quality next-token distribution. Since the $k$NN-LM relies on interpolating the next-token distribution from the baseline LM and that from the retriever, a lower quality retriever distribution can compromise the resulting next-token distribution and adversely affect the text generation performance.

We plot the ratio of Shannon entropy (Shannon, 2001) between the retriever distribution and that of the baseline LM distribution on the next token (with respect to the index of the token generated) and find that the retriever's entropy is increasing at a faster rate compared to that from the base-LM (Figure 3). [14] A higher entropy indicates lower level of confidence (closer to a uniform distribution over all tokens) and suggests that the retriever, when sampling long sequences, may be less reliable in identifying the high-quality tokens.

We hypothesize that the worsened reliability of the retriever over longer sampled sequences is likely a result of the *exposure bias* during text generation (i.e., at test-time, the retriever has to rely on model-generated queries that may contain artifacts or other distributional differences from human-written text). The retriever in $k$NN-LM

---

[14]Given a $|V|$-dimensional probability distribution $p$, the entropy is computed as: $H(p) = -\sum_{i=1}^{d} p_i \log(p_i)$

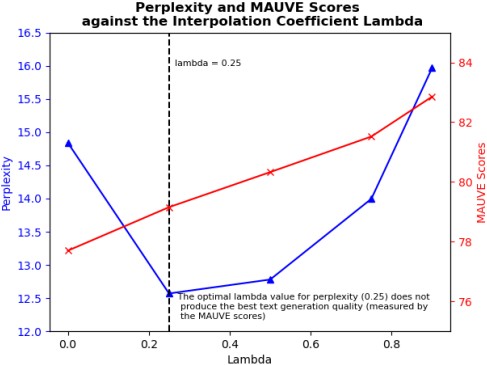

Figure 4: The interpolation coefficient $\lambda$ optimized for validation perplexity does not necessarily lead to the best text generation quality (measured by MAUVE).

is non-parametric since both the input prefix and the context from the datastore are encoded by the LM (without any additional retrieval parameters), which has been adapted to the training corpus of WikiText-103. However, during text generation, as the model iteratively samples tokens and appends them to the input prefix, the input context is more likely to deviate from those in the training corpus, hence, becomes more out-of-distribution and challenging for the retriever to accurately process.

## 6 Discussion

In addition to the limitations of interpolation-based LMs described in Section 5, we hypothesize that there are other potential factors that contribute to the shortcomings of the $k$NN-LM for text generation. Specifically, it is possible that the interpolation may impede the language models' ability for self-recovery, and also that integrating the retrieval distribution can potentially introduce additional burdens related to hyperparameter tuning, which may not be optimized for text generation. We discuss these potential issues here as they are interesting avenues to explore for future work.

**Retrieval interpolation may damage the self-recovery ability of LMs:** Language models exhibit some degree of self-recovery abilities (He et al., 2021b), i.e., they can regain fluency and coherence even after previously generating poor-quality tokens. This self-recovery capability is attributed to the LM's ability to pay close attention to recent context and ignore the long-range past context. However, we hypothesize that when interpolation-based LMs encounter artifacts (e.g., non-factual or disfluent text) in a distorted pre-

fix $\tilde{q}_t$, they may be less likely to recover, as the retrievals may further increase the probability of completions that resemble those artifacts. Furthermore, as we continuously sample and append tokens to the prefix, which the retriever uses as the query to construct $P_{KNN}(w_t|\tilde{q}_t)$, the retriever may encounter additional exposure bias as shown in Section 5.2, negatively impacting the quality of $P_{KNN}(w_t|\tilde{q}_t)$. Thus, even when the baseline LMs "recover" from distorted past context by producing a high-quality distribution over the next-token prediction $P_{LM}(w_t|\tilde{q}_t)$, the retriever may re-introduce the distortion by interpolating $P_{LM}(w_t|\tilde{q}_t)$ with $P_{KNN}(w_t|\tilde{q}_t)$.

**Hyperparameters introduced by $k$NN-LM are not optimized for text generation:** The $k$NN-LM introduces two important hyperparameters, namely the relative weight between the two distribution $\lambda$, as well as softmax temperature for the $k$NN distribution $\tau_{KNN}$. Recent work (Xu et al., 2023) highlights the significance of tuning $\tau_{KNN}$ for achieving optimal $k$NN-LM performance, as measured by perplexity. Similarly, we investigate the coefficient parameter $\lambda$, which plays a vital role as it controls the relative importance assigned to the $k$NN retriever and baseline LM. Existing works tune $\lambda$ by the perplexity on the validation set. However, from Figure 4, we observe that the $\lambda$ values that produce the lowest perplexity may not translate to the optimal value for text generation quality (measured by MAUVE). Instead of tuning $\lambda$ for optimizing perplexity, we may want to consider context-dependent $\lambda$ as in Drozdov et al. (2022) for generation (e.g., only use the retrieval distribution when it is very confident). Finally, interpolation may warrant the design of new decoding algorithms specialized for retrieval-augmented generation.

## 7 Conclusion

In this work, we show that despite the significant perplexity improvement brought by interpolation-based retrieval-augmented LMs such as $k$NN-LMs, such methods fail to improve the LMs' text generation performance. The text generation quality between $k$NN-LMs and baseline LMs without retrieval show no significant difference according to both automatic text generation evaluation metrics and human evaluation. Upon closer analysis, we identify flaws in using $k$NN-LMs to perform autoregressive text generation: the method only benefits a minority of token predictions, and the retriever's

quality deteriorates when generating long-form text. We hope our findings can inspire future research to design better training and inference methods so that the impressive improvement of $k$NN-LMs in perplexity can better be translated into gains in text generation quality.

## Ethics Statement

In this work, we investigate the text generation quality of language models. Language models can generate text that is harmful, offensive or unfaithful. We advise using caution when relying on language models to generate text and adopting post-processing strategies on the language-model generated text to remove undesirable content. Additionally, training large language models can bring significant energy cost. We hope that our analysis of the $k$NN-LM and future works on this topic may lead to more efficient method of using language models without the need to re-train such models.

## Limitations

Our work does not study all data, model, and evaluation configurations of interpolation-based LMs. Additionally, we focus on the 100M token datastore size, although kNN-LM can scale effectively to datastores of 3B words. Using a larger datastore may lead to further perplexity decreases, but we do not think this contradicts our finding that text generation degrades as retrieval quality does. We focus exclusively on interpolation-based LMs in this work, but similar issues for other retrieval-augmented LMs such as RETRO (Borgeaud et al., 2021) may also exist and be worth investigating further. Finally, our human evaluation does not specifically account for diversity, although some dimensions of this are captured by our automated metrics. Due to the overall low quality of text generated by LMs with and without retrieval, reading their outputs results in high cognitive burden on annotators, which might be ameliorated by using stronger LMs than GPT-2.

## Acknowledgements

We thank Zexuan Zhong and Danqi Chen for helpful discussion on TRIME and $k$NN-LM, and the UMass NLP group for feedback and discussion. We also thank the anonymous reviewers for their helpful comments.

This project was partially supported by awards IIS-2202506 and IIS-2046248 from the National Science Foundation (NSF). This research was also supported in part by a research gift from Adobe.

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

# A  Appendix

## A.1  Examples of $k$NN-LM hurting the inference of the next-token

We show examples where $k$NN-LM hurts the inference of the next-token in Table 4

## A.2  Human evaluation interface and examples

From our human evaluation, we show the interface for our evaluators in Fig 5 and also selected representative examples of evaluators' comments in Table 5.

## A.3  Models trained on PG-19 produce unhelpful artifacts

With retrieval from the datastore, the $k$NN-LM improves the perplexity on the validation dataset of the PG-19 marginally from 28.9 to 28.2 but does not improve the text generation quality. Both the baseline LM and the $k$NN-LM fine-tuned on the PG-19 dataset returns artifacts from the dataset (e.g. missing white-spaces and unnecessary line breaks), as shown in Table 6.

## A.4  Seq-Rep-$N$ of generated text from the baseline-LM and $k$NN-LM

Even though $k$NN-LM does not improve the text generation quality overall, we observe an improvement in lexical diversity (lower Seq-Rep-$N$) from $k$NN-LM on the WikiText-103 dataset in Table 7. However, this improvement in text diversity is obtained at the cost of Entity-F1 (a proxy for factuality).

| Context | Ground-truth | Most Probable Tokens from *base-LM* vs *kNN-LM* | Analysis |
|---|---|---|---|
| The lyrics were inspired by a story ...... To me , that 's the way a great rock ' n ' roll concert should be : a place where everyone comes together ... Maybe that 's the dream of all art : to break down the barriers and the divisions between | "people" *base-LM* probability: 0.26 *kNN-LM* probability: 0.23 | *base-LM*: "the"(0.20), "us"(0.09), "art"(0.03), "rock"(0.02) *kNN-LM*: "the"(0.23), "us"(0.07), "good"(0.02), 'art"(0.02) | In this example the *base-LM* predicts the ground-truth **noun** token "people" with the highest probability of all tokens (0.26). However, after interpolating with the retrieval distribution, the *kNN-LM* decreases the probability of the ground-truth token. |
| Richmond finished the 1984 season 12th in points , with 11 ...... In the Busch Series , he qualified at the pole position in the two races he entered , and won the Charlotte race . Richmond joined Hendrick Motorsports in 1986 , where he teamed up with veteran crew chief Harry Hyde . It took the team until the middle of the season' | "to" *base-LM* probability: 0.78 *kNN-LM* probability: 0.64 | *base-LM*: ","(0.07), "for"(0.03), "when"(0.02), 'that"(0.02) *kNN-LM*: ","(0.10), "for"(0.06), "."(0.04), "and"(0.02) | The ground-truth token to be predicted is the **preposition "the"**, which the *base-LM* correctly predicts with very high probability. However, the *kNN-LM* decreases the probability of the ground-truth token significantly compared to the *base-LM*. |

Table 4: Examples where *k*NN-LM hurts the inference of next-token (with different part-of-speech such as noun and preposition) by predicting of the gold token with a lower probability compared to the base-LM

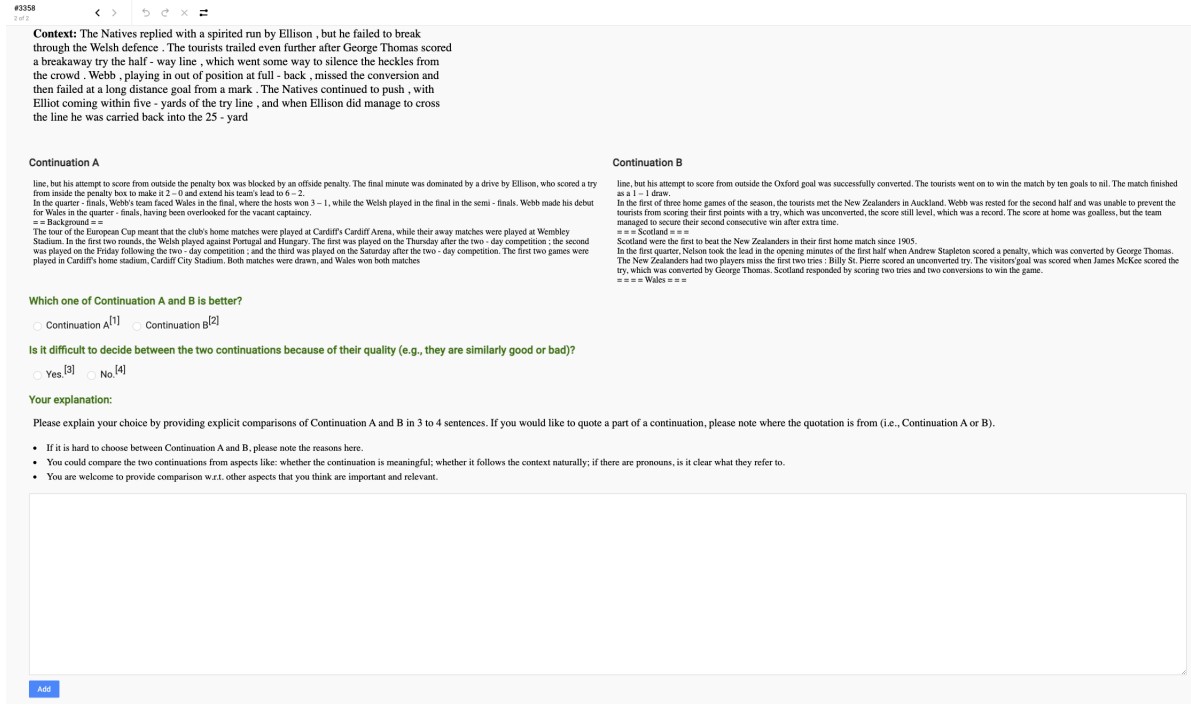

Figure 5: The interface of the human evaluation. Each task consists of a context text, two continuations, two choices, and a free-form justification text box.

| Context | Continuation | Comment | Error |
|---|---|---|---|
| [. . .] = = = Shootdown = = = A group of ZIPRA guerrillas, armed with a Strela-2 launcher, waited in the bush beneath Flight 825's flightpath, and | **Continuation A—kNN-LM** two soldiers opened fire on the guerrillas. [. . .] The ZIPRA and ZIPRA units exchanged fire, and several ZIPRA members were shot. After the fire was over, the ZIPRA and ZIPRA guerrillas returned fire and began shooting back, and one of the ZIPRA unsuccessfully attacked a ZAPU base at the head of Flight 825, forcing the major to retreat. = = = = ZAPU retreat = = = = = = = = ZAPU retreat = = = = [. . .] The ZIPRA moved up the hill and captured several ZIPRA members, forcing them to withdraw. [. . .] | Both of these continuations have problems. I chose A because its problems seems to influence the meaning less than the problems in B. [. . .] Continuation A has some repeated words and in a few spots ZIPRA and ZAPU seem to get mixed up, leading to sentences like this: "The ZIPRA moved up the hill and captured several ZIPRA members, forcing them to withdraw." It was easier to recover from those confusions than from the ones in Continuation B. | Repetitive content; contradiction |
| Hu was born in \<unk\> County, Anhui Province in 1584 or early 1585. [. . .] By 1619, Hu had moved to Nanjing where he lived with his wife Wu. | **Continuation B—GPT-2** During this time he also became a leading scholar of the court, a position he held until 1624, [. . .]. His last year at the academy was 1715. In 1725, Hu was appointed secretary of the government [. . .]. In 1727, he was appointed as vice-president of the Academy, and he remained the academy's secretary until 1742, [. . .]. | I selected that this was a difficult decision [. . .]. However, Continuation B is an impossible choice as it has Hu being born in 1584 and still being alive in 1725 which of course is impossible. I was left with process of elimination in my opinion. | Illogical; wrong chronological details |
| The lanes of US 75, which are on the bottom level, are 20 feet (6m) below ground level, giving the structure a total height of 140 feet from bottom to top As part of the project, I-635 was widened to include four dedicated high-occupancy vehicle lanes (HOV lanes) that are barrier-separated and reversible. The roads on the five levels are: Level I: US 75, an eight lane highway, four lanes going each way; Level II: The junction of the two six-lane frontage roads, each having three through | **Continuation A—kNN-LM** traffic lanes with one lane in each direction. Level III : In the northbound direction, US 75 curves to the northeast and US 75 curves to the northwest to follow a more southerly routing. The northbound lanes are two lanes in each direction, while the southbound lanes are two lanes. [. . .] = = Major intersections = = [. . .] = Battle of the Kalka River = The Battle of the Kalka River (Polish : Kalka, pronounced [ˈkaʊ], or [ˈkaʊˈmaʊ]) was a decisive battle of the 15th and 16th centuries of the 15th century in Austrian occupied Poland. The battle was fought in the middle of the Russo | Continuation A gives more information and is more specific than B. It has some lack of clarity – Level III has the road curving both northeast and northwest – but overall it's more informative than just "the intersection of two frontage roads." I would have ruled out A because of the Battle of the Kalka River section at the bottom. It doesn't really have a place in this text. But B also has the Battle of the Falkland Islands, so that's a wash. A it is. | Abrupt topic change; illogical |
| (Same as above) | **Continuation B—GPT-2** traffic lanes. Level III : The intersection of the two six-lane frontage roads, each having four through traffic lanes. Level IV: The intersection of the two six-lane frontage roads, each having two through traffic lanes. Level V: The intersection of the two six-lane frontage roads, each having two through traffic lanes = = = Construction = = = The project was approved in July 1997 and completed on December 17, 1997, with the groundbreaking ceremony taking place at the State University of New York (SUNY) campus in New York City. [. . .] = = = Major intersections = = = [. . .] = Battle of the Falkland Islands = | (Same as above) | Abrupt topic change; repetitive content |

Table 5: Selected representative examples with evaluators' comments. The error categories in the last column are ours. Contents are shortened for the sake of space.

| Context | baseline-LM-generated text | kNN-LM-generated text |
|---|---|---|
| For my part,something within me answered to Kandinsky's art the first time I met with it. There was no question of looking for representation; harmony had been | in my brain, and I had been working at it ever since.That is why I was able to tell Kandinsky what was going on within me. Idid not need to be told about his melody. I knew that he was doing itwith the greatest energy and energy and energy and energy. | in my brain, and I had been working, like theothers, in that direction for some years. It was this which prompted me totry for a musical conception. I wanted to be able to make music tosuit the moods and the contrasting circumstances of the moment, |
| Before going toMaria, they insist on doing the honors, and showing the objectsinterest the vicinity. So they take us to their barrack, a large-house, and thence to ïhe front.Ťo the latter spot our coachmandriving | the family out of the house, and we go through the sameprocess with them. Our next station is in the heart of San Mateo, where theyhave a ḧousewith a room on the outside, and ... and afloor and a ceiling. | the family out of the house, and, with a chuckle, heexplains the advantages of boarding-schools in the United States, aswell as of boarding-schools in France and Spain. I am reminded of thisexception, and feel that there is a difference in the methods of boardingschools in the two countries. |

Table 6: Both the baseline LM and kNN-LM generate text that consists of artifacts from the dataset, e.g. missing white-spaces between tokens

| Model | Seq-Rep-1 | Seq-Rep-2 | Seq-Rep-3 | Seq-Rep-4 |
|---|---|---|---|---|
| *kNN-LM with and without retrieval from Alon et al. (2022)* | | | | |
| GPT-2 small *(no retrieval)* | 56.7 | 26.6 | 15.1 | 9.65 |
| GPT-2 small *(+ retrieval)* | 53.3 | 22.5 | 11.6 | 6.73 |

Table 7: Even though the $k$NN-LM does not improve the overall text generation quality, we observe higher lexical diversity (lower Seq-Rep-$N$) in the $k$NN-LM-generated text, on the WikiText-103 dataset, using the GPT2-small model as the baseline LM.

