# OpenReview forum: "$k$NN-LM Does Not Improve Open-ended Text Generation"
_EMNLP/2023/Conference — EMNLP 2023 Main_

### Official Review · Reviewer_FiTG · 2023-08-03

**Soundness:** 4

**Excitement:**

4: Strong: This paper deepens the understanding of some phenomenon or lowers the barriers to an existing research direction.

**Paper Topic And Main Contributions:**

This paper investigates the impact of interpolation-based retrieval-augmented LMs on open-ended text generation by looking at two different kNN-LM models on different datasets, concluding that these interpolated LMs cannot improve open-ended text generation task. Extensive analysis and discussions are conducted to explain the potential reasons behind the observations.

The main contribution is that this work provides different insights of retrieval-augmented LMs for open-ended text generation compared to previous works which shows improvements on perplexity and classification-related tasks.

**Questions For The Authors:**

Refer to the questions in Reasons to reject.


**Reasons To Accept:**

The main contribution is that this work provides different insights of retrieval-augmented LMs for open-ended text generation compared to previous works which shows improvements on perplexity and classification-related tasks.


**Reasons To Reject:**

The evaluation of open-ended text generation is a difficult task, the evaluation results of automatic metrics and human evaluation in this paper might not be plausible. For example, the quality of human evaluation replies on many important aspects, such as the quality of evaluators, the aspects to be checked. From Figure 1 we can see that Rate1 and Rate2 are more confident on the kNN-LM while Rate3 is more for the GPT-2, which is very strange. Furthermore, the human evaluation looks at grammatical, fluent, consistent, and logical, we need to look at how each model performs at each aspect, e.g. would kNN-LM performs better than GPT-2 in terms of consistent and logical?

**Reproducibility:**

5: Could easily reproduce the results.

**Reviewer Confidence:**

4: Quite sure. I tried to check the important points carefully. It's unlikely, though conceivable, that I missed something that should affect my ratings.

---

> ### Author Rebuttal · Authors · 2023-08-29
>
> We thank reviewer FiTG for recognizing the value of our experiments and for their insightful suggestions!
>
> > It is strange that rater 1 and Rater 2 more confident on KNNLM but Rater 3 more confident for GPT2
>
> Different raters show different preference (and level of confidence) in the comparison between the base-LMs and the KNN-LMs mainly due to the fact that the quality of text generated from these models is very difficult to distinguish (examples in Table 5 & 6 of appendix). This observation underscores the fact that kNN-LM does not bring significant improvement in text generation despite improved perplexity. As mentioned in our response to reviewer qeVJ, we will obtain additional human annotations for the next version of our paper.
>
> > Need to look at how each model performs at each aspect ---- does knnlm outperform GPT2 in terms of consistency / logic?
>
> Contrary to the improvement in perplexity, we find that kNN-LM does not bring significant improvement in any of the four aspects of our human evaluation (grammatical, fluent, consistent, and logical). We will include more discussion in the next version of the paper.

---

### Official Review · Reviewer_qeVJ · 2023-08-06

**Typos Grammar Style And Presentation Improvements:** Line 632
**Soundness:** 4

**Excitement:**

4: Strong: This paper deepens the understanding of some phenomenon or lowers the barriers to an existing research direction.

**Paper Topic And Main Contributions:**

This work explore the performance of  interpolation-based retrieval-augmented language models on text generation. Although these models might have advantages in other practical applications, the findings from the evaluation results show that they actually do not produce texts in higher quality even they have generally lower perplexity as reported in previous studies.



**Questions For The Authors:**

Question A: according the phenomenons revealed in section 5, how should the future retrieval-based LMs improved?

**Reasons To Accept:**

- The authors have done thorough evaluation on interpolation baselines and provide useful insight that models like kNN-LM improve the perplexity only because they largely improve prediction accuracy on a small subset of tokens, and the quality of such retrieval-based LMs deteriorates over longer sequences.

- Designs good analytical experiments on the negative results on the perplexities
  - Exhibit the increases brought by the nearest neighbour search by POS tags -- helpful to learn that which kinds of tokens should be further improved
  - Using the ratio of Shannon entropy to show the confidence of the prediction.
- As an analysis paper, this work provides comprehensive baselines and automatic metrics from various perspectives (MAUVE, RankGen, GPT-3 perplexity, Entity-F1, Seq-Rep-) and human evaluation.

**Reasons To Reject:**

- Scope is limited to interpolation-based methods like kNN-LM, findings may not generalise to other retrieval augmented LMs.
- Human evaluation is limited with only 3 raters of 45 examples. More data could strengthen conclusions.

**Reproducibility:**

4: Could mostly reproduce the results, but there may be some variation because of sample variance or minor variations in their interpretation of the protocol or method.

**Reviewer Confidence:**

4: Quite sure. I tried to check the important points carefully. It's unlikely, though conceivable, that I missed something that should affect my ratings.

---

> ### Author Rebuttal · Authors · 2023-08-29
>
> We thank reviewer qeVJ for recognizing the value of our analysis and for their insightful reviews.
>
> > Results may not generalize to other retrieval-augmented LMs except interpolation-styled retrieval LMs:
>
> We limit the scope of our work to interpolation-styled retrieval-LMs due to their popularity and effectiveness in improving perplexity, in addition to the open-source availability of pretrained checkpoints and datastores (see response to reviewer ofKA for more details). While other types / architectures of retrieval language models may exhibit different behaviors for open-ended text generation, we hope our work can inspire future research to include **quality of generated text** in their evaluation protocol instead of just perplexity.
>
>
> > Human evaluation is limited with only 3 raters, 45 examples. Need more data points.
>
> Instead of conducting crowd-sourced evaluation, we carefully select and hire three annotators who are professional English teachers/editors and are experienced in text generation evaluation, following the recommendations from Karpinska et al [1]. That said, we acknowledge the reviewer’s concern and will hire these same annotators to provide more judgments for the next version of the paper.
>
>
> ### [Question]: According to the phenomenons revealed in section 5, how should the future retrieval-based LMs be improved?
>
> We believe future work can improve text generation from similar retrieval language models by investigating the following directions.
> 1. Improving the retrieval quality (such as ensembling the retrieval process with pre-trained dense-retrievers),
> 2. Using modified decoding strategies (such as filtering or re-calibrating the retrieval distribution)
> 3. Adopting an adaptive retrieval strategy (learning when to retrieve rather than always retrieving at every token)
>
> ### References
> 1. *The Perils of Using Mechanical Turk to Evaluate Open-Ended Text Generation*. EMNLP 2021.

---

### Official Review · Reviewer_ofKA · 2023-08-12

**Soundness:** 4

**Excitement:**

4: Strong: This paper deepens the understanding of some phenomenon or lowers the barriers to an existing research direction.

**Paper Topic And Main Contributions:**

This paper conduct relatively comprehensive experiments to deeply study and analyze the kNN-LM, which is an interpolation-based retrieval-augmented language model. Besides, this paper finds that interpolating with a retrieval distribution actually increases perplexity compared to the baseline LM for the majority of tokens in the WikiText-103 test set, which is a little counter-intuitive, because the overall perplexity is lower.

**Reasons To Accept:**

The experimental findings of this paper are relatively novel, and are beneficial for the research of natural language generation.

**Reasons To Reject:**

This paper does not analyze more advanced models thar are based on kNN-LM. kNN-LM is a hot research topic in recent years, and there are many improvements over it, but this paper only uses the original model and the TRIME.

**Reproducibility:**

4: Could mostly reproduce the results, but there may be some variation because of sample variance or minor variations in their interpretation of the protocol or method.

**Reviewer Confidence:**

4: Quite sure. I tried to check the important points carefully. It's unlikely, though conceivable, that I missed something that should affect my ratings.

---

> ### Author Rebuttal · Authors · 2023-08-29
>
> We thank reviewer ofKA for recognizing the novelty of our experimental findings and for their insightful comments.
>
>
> ### Why focus on the original kNN-LM and TRIME?
>
> In this work, we focus on kNN-LM and TRIME for the following reasons, which we will clarify in the next version of the paper. First, both of these methods are interpolation-based retrieval-augmented LMs, as opposed to methods that feed retrieved text into the LM input or fuse it at other layers. Interpolation-based methods such as the kNN-LM are convenient because they are easily applicable to any pretrained LM without any further fine-tuning; as such, the kNN-LM has been cited over 400 times since its publication. We acknowledge that other retrieval LMs such as RETRO likely yield better performance, but these models are not open-sourced and it’s not possible to easily adapt existing pretrained LMs to simulate them.
>
> Meanwhile, both the kNN-LM and TRIME papers include publicly-available model checkpoints, datastores, and codebases, which makes controlled comparisons feasible.
>
> Additionally, the TRIME model introduces substantial modifications to the conventional kNN-LM by training the retrieval process. This design decision leads to significant improvements over the kNN-LM, and in some ways offers a more comparable analogue to other retrieval paradigms such as RETRO. However, as our paper demonstrates, making the retrieval process trainable does not translate to improvements in text generation quality. We believe this finding is novel and surprising, and thus of importance to the growing body of work on retrieval-augmented language modeling.
>
>
> ### What about other variations of kNN-LM?
> We agree with reviewer ofKA that there are other interesting variations of kNN-LM that we did not study in this paper. We chose not to include these in our paper with the following reasons.
> 1. *Efficient Nearest Neighbor Language Models*. EMNLP 2021 (focusing on improvement in inference speed instead of language modeling quality)
> 2. *Nearest Neighbor Machine Translation*. ICLR 2021 (focusing on performing machine translation instead of open-ended text generation)
> 3. *Neuro-Symbolic Language Modeling with Automaton-augmented Retrieval*. ICML 2022 (requiring non-trivial modification to perform text generation through sampling)
>
> We hope our experiments can stimulate further research to follow our evaluation framework when evaluating future retrieval-based LM variations.

---

### Meta-Review · Area_Chair_Jvub · 2023-09-17

**Recommendation:** 5

**Metareview:**

The paper evaluates the generation quality of interpolation-based retrieval-augmented language models, exemplified by kNN-LM, which interpolates the predictions of the LM with the predictions based on the most relevant retrievals from the prefix. They find that while such methods improve overall perplexity, the do not improve the quality of open-ended text generation, using both automatic metrics and human evaluations. They provide analyses and plausible explanation of the perplexity decrease reported in previous research. The paper is well written and the experiments well constructed, with insightful analysis of the existing interpolation-based kNN-LM model that can guide better evaluation of similar models in the future.

---

### Decision · Program_Chairs · 2023-10-07

**Decision:**

Accept-Main

**Comment:**

The paper evaluates the generation quality of interpolation-based retrieval-augmented language models, exemplified by kNN-LM, which interpolates the predictions of the LM with the predictions based on the most relevant retrievals from the prefix. They find that while such methods improve overall perplexity, the do not improve the quality of open-ended text generation, using both automatic metrics and human evaluations. They provide analyses and plausible explanation of the perplexity decrease reported in previous research. The paper is well written and the experiments well constructed, with insightful analysis of the existing interpolation-based kNN-LM model that can guide better evaluation of similar models in the future.